# Hypofractionated Radiotherapy in Intermediate-Risk Prostate Cancer Patients: Long-Term Results

**DOI:** 10.3390/jcm11164783

**Published:** 2022-08-16

**Authors:** Maurizio Valeriani, Mario Di Staso, Giuseppe Facondo, Gianluca Vullo, Vitaliana De Sanctis, Giovanni Luca Gravina, Milena di Genesio Pagliuca, Mattia Falchetto Osti, Pierluigi Bonfili

**Affiliations:** 1Department of Medicine and Surgery and Translational Medicine, Sapienza University of Rome, Radiotherapy Oncology, St Andrea Hospital, 00189 Rome, Italy; 2Radiotherapy Oncology Unit, University of L’Aquila, St Salvatore Hospital, 67100 L’Aquila, Italy; 3Radiotherapy Oncology Unit, Giuseppe Mazzini Hospital, 64100 Teramo, Italy; 4Radiotherapy Oncology Unit, Pescara Hospital, 65121 Pescara, Italy

**Keywords:** intermediate-risk prostate cancer, hypofractionated radiotherapy, prognostic factors, 3D-CRT, PSA

## Abstract

Background: To evaluate outcomes in terms of survival and toxicity in a series of intermediate-risk prostate cancer (PCa) patients treated with hypofractionated radiotherapy (HyRT) + hormonal therapy (HT) with or without image guidance (IGRT) and to investigate the impact of different variables. Methods: This is a multi-centric study. From January 2005 to December 2019, we treated 313 intermediate-risk PCa patients (T2b–T2c, Gleason score 7, or pre-treatment PSA 10 to 20 ng/mL) with HyRT. Patients received 54.75 Gy in 15 fractions in 5 weeks plus 9 months of neo-adjuvant, concomitant, and adjuvant HT with or without IGRT. Results: Median follow-up was 91.6 months (range 5.1–167.8 months). Median OS was not reached, and the 8- and 10-year OS was 81.9% and 72.4%, respectively. Median CSS was not reached, and the 8- and 10-year CSS was 97.9% and 94.5%, respectively. PSA at first follow-up <0.8 ng/mL was significantly related to better oncological outcomes (CSS, bRFS, LRFS, cPFS, and MFS) in both univariate and multivariate analysis. After Propensity Score matching, grade 2–3 acute and cumulative late GU (*p* = 0.153 and *p* = 0.581, respectively) and GI (*p* = 0.196 and *p* = 0.925, respectively) toxicity were not statistically different in patients treated with or without IGRT. Conclusions: HyRT is effective and safe regardless of the use of IGRT. PSA at first follow-up is an easily accessible prognostic factor that may help the clinicians to identify patients who require a treatment intensification.

## 1. Introduction

Local treatments such as surgery and radiotherapy (RT) are the standard of care in the management of localized prostate cancer (PCa) [1]. High-dose conventional fractionated radiotherapy leads to high rate of disease control with a low rate of high-grade toxicities, but the overall treatment time is about 7–8 weeks [2,3].

Generally, prostate cancer is characterized by a low proliferation rate and a low α/β ratio in the linear-quadratic model. Radiobiological studies have determined the α/β ratio for prostate cancer to be approximately 1.5 Gy, unlike most tumours that have a higher α/β ratio (10 Gy on average), making this cancer particularly sensitive to high doses per fraction [4,5,6]. In this case, hypofractionated RT (HyRT) may be advantageous for tailoring the treatment in order to maintain a high rate of disease control, a low rate of late toxicity, and a reduction in the number of patient accesses to the hospital with obvious advantages.

Three non-inferiority phase 3 randomized controlled trials have confirmed the safety and efficacy of moderate hypofractionation (2.5–3.0 Gy per fraction) [7,8,9].

In intermediate and high-risk patients, the addition of hormonal therapy (HT) to RT has demonstrated an advantage [10,11,12,13] with respect to RT alone. In fact, neo-adjuvant and concomitant short-course HT in combination with RT is usually considered as a standard treatment for intermediate-risk PCa. The aim of this study was to evaluate efficacy, toxicity, and prognostic factors in a series of intermediate-risk prostate cancer patients treated with 15 fractions of HyRT + HT with or without image-guided radiotherapy (IGRT).

## 2. Materials and Methods

### 2.1. Patients’ Characteristics

Three-hundred and thirteen consecutive patients with intermediate-risk prostate cancer, by the definition of NCCN guidelines (T2b–T2c, Gleason score 7, or pre-treatment PSA 10 to 20 ng/mL), were treated between January 2005 and December 2019. After a transrectal ultrasound (TRUS)-guided biopsies, all patients had histologically confirmed prostate cancer. A complete history, physical examination with digital rectal examination, blood tests including PSA level, contrast-enhanced total body computed tomography, and bone scan were obtained before treatment. Local staging was assessed with pelvic multiparametric magnetic resonance imaging (MRI) including perfusion study, T2, and diffusion-weighted imaging, in all cases. Patients were divided into two prognostic groups: (1) favorable prognostic group (FG) if they had PSA 10–20 ng/mL or T2b–T2c and Gleason score ≤ 3 + 4 = 7, and (2) unfavorable prognostic group (UG) if they had PSA 10–20 ng/mL and T2b–T2c and/or Gleason score 4 + 3 = 7 [14].

After the approval of our internal review board (IRB), all patients signed a written consent before the start of treatment. The data were prospectively collected and retrospectively analyzed.

The median age at diagnosis was 74 years (range 48–88). One-hundred and eighty-nine patients (60.4%) were classified as FG and 124 (39.6%) as UG. Sixty-eight patients (21.8%) had T1c clinical stage, 92 (29.4%) with T2a, 78 (24.9%) with T2b, and 75 (23.9%) with T2c. The median PSA at diagnosis was 8.15 ng/mL (range 2.6 to 19.9 ng/mL). Ninety-one patients (29.1%) had a Gleason score of 6 (3 + 3), 133 (42.5%) of 7 (3 + 4), and the remaining 89 (28.4%) of 7 (4 + 3). Patients’ characteristics are shown in Table 1.

Patients underwent a planning CT scan, with 2.5 mm slices from the anal verge to the L5–S1 interface, in the supine position with a footrest device. Before planning CT scan, patients were asked to self-administer a mini enema to empty the rectum. Additionally, they were instructed to empty their bladder and drink 500 mL of water about 30 min prior to the procedure. T2 weighted, early perfusion, and apparent diffusion coefficient (ADC) map MRI series were fused with planning CT to better delineate the clinical target volume (CTV) when available. CTV1 included the prostate plus seminal vesicles and CTV2 included the prostate alone. Planning target volumes (PTV1 and PTV2, respectively) were generated with an 8 mm margin in all directions, except posteriorly, where a 6 mm expansion was adopted in 136 patients (43.5%) treated without IGRT. A 5 mm expansion in all directions was used in the other 177 patients (56.5%) as daily kv cone beam CT (CBCT) was used to verify the patient position. The whole rectum from the anus to the sigmoid flexure, bladder, femoral heads, and penile bulb were contoured as organs at risk. A 3D conformal radiotherapy (3D-CRT) plan was performed with five coplanar fields and a 15 MV photon linear accelerator was used to deliver the treatment. PTV1 received 43.8 Gy in 12 fractions and PTV2 received 3 additional fractions of 3.65 Gy for a total of 54.75 Gy in 15 fractions, three times a week in order to avoid an excess of acute toxicity. According to the linear quadratic model, this RT regimen is biologically equivalent to 80.5 Gy in 2 Gy fractions assuming a α/β ratio of 1.5 Gy. This regimen is also equivalent to 72.8 Gy in 2 Gy fractions assuming a α/β ratio of 3 Gy for late-responding tissue. With the aim to maintain a low rate of acute toxicity and the consequential late damage, we decided to maintain the overall treatment time not below 5 weeks [15]. Dose–volume constraints were as follows: V45 < 35% and V52 < 25% for the rectum, and V40 < 50% for the bladder. Patient position was verified using an electronic portal-imaging device (EPID) in the non-IGRT group, while the IGRT group underwent daily CBCT. Neo-adjuvant, concomitant, and adjuvant hormonal therapy (HT) was started 3 months before RT for a total duration of 9 months for hormonal treatment. HT consisted of anti-androgen or LHRH-analogue according to the treating physician’s preference and was administered to all patients. Ninety-seven patients (31.0%) were treated with anti-androgen and 216 (69.0%) with LHRH analogous.

### 2.2. Toxicity and Follow-Up

The first follow-up was performed 30–45 days after the end of radiotherapy, then every 3 months for the first year and every 6 months afterwards. Toxicities were evaluated according to the Radiation Therapy Oncology Group (RTOG) scale for acute and late adverse effects at each follow-up [16].

### 2.3. Statistical Analysis

Receiver operating characteristic (ROC) curves were used to find the cut-off values for continuous variables. Biochemical failure was defined as the PSA nadir after RT + 2 ng/mL according to the Phoenix criteria [17]. Local recurrence was considered as the relapse of the tumor in the prostate, seminal vesicles, or loco-regional lymph nodes at PET scan with choline, MRI, or biopsy. The median follow-up was calculated using the “reverse” Kaplan–Meyer method [18]. Overall survival (OS), cancer-specific survival (CSS), biochemical recurrence-free survival (bRFS), the local recurrence free survival (LRFS), clinical progression-free survival (c-PFS), and metastasis-free survival (MFS) were calculated after the end of RT until the event or the last follow-up if the event did not occur. The curves were generated using the Kaplan–Meier method analyzing the following variables: age (<70 vs. ≥70 years), hormonal therapy (LHRH analogue vs. anti-androgen), IGRT (yes vs. no), PSA pre-RT (≤0.7 vs. >0.7 ng/mL), PSA at first follow-up (≤0.8 vs. >0.8 ng/mL), and risk group (FG vs. UG). The Cox proportional hazards model was used for multivariate analysis including age, risk group, IGRT, PSA pre-RT, and PSA at first follow-up. Patients treated with IGRT were matched to patients treated without IGRT, via propensity score matching analysis. The following variables were included in the model: age, PSA at diagnosis, Gleason score, tumor stage, and hormonal therapy. The two groups were matched together using a 0.2 width caliper of the propensity score standard deviations to minimize confounding bias. Owing to the paired nature of the matched data, which is 1:1, we used a cluster variable to identify the matched pairs. Chi-square test was used to compare characteristics of patients treated with or without IGRT before and after propensity score matching analysis and to compare grade 2–3 acute and cumulative late toxicities in patients treated with or without IGRT. Statistical analyses were performed with SPSS statistical software for Macintosh version 25.0 (SPSS, Inc., Chicago, IL, USA). A value of *p* ≤ 0.05 was considered statistically significant.

## 3. Results

Propensity score matching was performed identifying 224 patients (112 cases treated with IGRT and 112 without) with overlapping characteristics for survival analysis (Table 2). Median actuarial follow-up was 91.6 months (95% c.i. 81.8–101.4, range 5.1–167.8 months). Median follow-up for surviving patients was 82.3 months (range 24.5–167.8 months). Sixty-four patients (20.4) died—52 (16.6%) from intercurrent disease and 12 (3.8%) from PCa. Median OS was not reached, and the 8- and 10-year OS was 81.9% and 72.4%, respectively. Median CSS was not reached, and the 8- and 10-year CSS was 97.9% and 94.5%, respectively (Figure 1). There were no differences between FG and UG in terms of CSS (8-year OS FG 99.1% vs. 95.8% UG, *p* = 0.962). Patients treated without IGRT presented an 8-year CSS of 99.2% versus 95.7% in patients treated with IGRT (*p* = 0.361). Hormonal therapy (HT) (*p* = 0.078) and PSA pre-RT (*p* = 0.145) did not influence CSS. Moreover, age (8-year CSS for patients with <70 years 93.8% vs. 98.0% in patients with ≥70 years, *p* = 0.087) did not influence CSS. Only PSA at first follow-up (8-year CSS for patients with a PSA value ≤ 0.8 ng/mL at first follow-up was 98.9% vs. 88.6% in patients with PSA > 0.8 ng/mL, *p* <0.001) (Figure 2) significantly influenced CSS. Multivariate analysis confirmed statistical significance only for PSA at first follow-up (*p* = 0.030) for CSS. After PS matching, both univariate and multivariate analysis confirmed PSA at first follow-up as a prognostic factor (*p* = 0.001 and *p* = 0.027, respectively).

Forty-one patients (13.1%) developed biochemical recurrence after a median follow-up of 41.4 months (range 10.9–135.8 months). Median bRFS was not reached, and the 8- and 10-year bRFS was 85.9% and 78.8%, respectively. In the univariate analysis, the use of IGRT did not influence bRFS both before and after PS matching (*p* = 0.581 and *p* = 0.249, respectively), whereas PSA at first follow-up (*p* < 0.001 before and *p* = 0.011 after PS matching) was the only statistically significant prognostic factor. Moreover, in the multivariate analysis, PSA at first follow-up was the only factor that influenced bPFS.

Twenty-three patients (7.3%) developed local recurrence after a median follow-up of 69.4 months (range 13.2–111.6 months). Median LRFS was not reached, and the 8- and 10-year LRFS was 90.4% and 86.5%, respectively. Univariate analysis did not show a significant difference in terms of the use of IGRT both before and after PS matching (*p* = 0.275 and *p* = 0.467, respectively). The only factor that influenced LPFS was the PSA at first follow-up (*p* = 0.014 before PS matching), but this result was not confirmed after PS matching (*p* = 0.103). Multivariate analysis confirmed these results.

Twenty-six patients (8.3%) developed clinical progression after a median follow up of 60.7 months (range 12.4–111.6 months). Median cPFS was not reached, and the 8- and 10-year cPFS was 89.5% and 85.6%, respectively. Among the 26 patients with clinical recurrence, 12 (46.1%) had local recurrence, 3 (11.5) developed distant metastases, and 11 (42.4%) had both local recurrence and distant metastases.

Patients in the FG presented an 8-year cPFS of 93.1%, whereas that of those in the UG group was 83.1% (*p* = 0.042). In addition, PSA at first follow-up was statistically different in terms of cPFS (8-year cRFS for patients with a PSA value < 0.8 ng/mL was 91.0% vs. 73.7% in patients with PSA ≥ 0.8 ng/mL, *p* = 0.005). Age (*p* = 0.750), HT (*p* = 0.659), IGRT (*p* = 0.443), and PSA pre-RT (*p* = 0.665) did not statistically influence cRFS. After PS matching, IGRT (*p* = 0.835), age (*p* = 0.868), HT (*p* = 0.868), PSA pre-RT (*p* = 0.940), and risk group (*p* = 0.884) did not influence cRFS, whereas PSA at first follow-up (*p* = 0.026) was statistically correlated with cRFS.

Multivariate analysis showed statistical significance for FG versus UG (*p* = 0.047) and PSA at first follow-up (*p* = 0.010) before PS matching and for PSA at first follow-up after PS matching (*p* = 0.037).

Fourteen patients (4.5%) developed distant metastases after a median follow-up of 65.2 months (range 12.4–114.8 months).

Median MFS was not reached, and the 8- and 10-year MFS was 96.0% and 91.4%, respectively. PSA at first follow-up was statistically different in terms of MFS (*p* = 0.004 and *p* = 0.003 before and after PS matching, respectively). Age, risk group, OT, IGRT, and PSA pre-RT did not statistically influence MFS both before and after PS matching. Multivariate analysis confirmed statistical significant differences for PSA at first follow-up. The data are extensively reported in Table 3 and Table 4.

## 4. Toxicities

The dose constraints for the rectum and bladder were respected in all patients and the dose to the targets varied between 95% and 107% of the prescription dose.

Overall, the treatment was well tolerated. During treatment, 126 patients (40.2%) presented grade 1 acute genitourinary (GU) toxicity, 16 (5.1%) presented grade 2, and 2 (0.6%) presented grade 3; 62 patients (19.8%) presented grade 1 acute gastro-intestinal (GI) toxicity and 15 (4.8%) presented grade 2.

At first follow-up, 74 patients (23.6%) presented grade 1 acute GU toxicity and 10 (3.2%) presented grade 2–3; 19 patients (6.0%) presented grade 1 acute GI toxicity and 2 (0.6%) presented grade 2.

At last follow-up, 35 patients (11.2%) presented grade 1 late GU toxicity and 13 (4.2%) presented grade 2–3; 17 patients (5.4%) presented grade 1 late GI toxicity and 5 (1.6%) presented grade 2–3.

Cumulative incidence of late toxicities was as follows: 69 patients (22.0%) presented during follow-up grade 1 late GU toxicity and 17 (5.4%) presented grade 2–3; 30 patients (9.6%) presented grade 1 late GI toxicity and 9 (2.9%) presented grade 2–3.

Grade 2–3 acute and cumulative late GU toxicity were 7.9% and 6.2%, respectively, in patients treated with IGRT and 2.9% and 5.1%, respectively, in patients treated without IGRT (*p* = 0.061 and *p* = 0.688, respectively).

Grade 2–3 acute and cumulative late GI toxicity were 6.8% and 2.8%, respectively, in patients treated with IGRT and 2.2% and 2.0%, respectively, in patients treated without IGRT (*p* = 0.060 and *p* = 0.725, respectively).

After PS matching, grade 2–3 acute and cumulative late GU (*p* = 0.153 and *p* = 0.581, respectively) and GI (*p* = 0.196 and *p* = 0.925, respectively) toxicity were not statistically different in patients treated with or without IGRT.

## 5. Discussions

In the last decades, several randomized trials have compared moderate HyRT (dose per fraction of 2.4–3.5 Gy) to conventional fractionation (dose per fraction of 1.8–2 Gy) in prostate cancer patients, demonstrating that HyRT is non-inferior in terms of outcomes and safety with respect to conventional fractionation [7,8,19,20,21,22,23].

Our study analyzed oncological outcomes and prognostic factors in a group of 313 patients with intermediate-risk PCa treated with HyRT and HT. The peculiarity of our study is the use of a hypofractionated schedule with 15 fractions of 3.65 Gy three times a week and, to the best of our knowledge, it is the only study in the literature that uses this treatment schedule and a long follow-up. After a median follow-up of 91.6 months, the 10-year OS and CSS for the entire cohort were 72.4% and 94.5%, respectively, and bRFS was 78.8%. Forty-one (13.1%) patients developed a biochemical recurrence and 26 (8.3) out of these also had clinical detectable relapse (12 (5.3%) only local recurrence, 3 (1.3%) only distant metastases, and 11 (4.9%) both local recurrence and distant metastases).

There is emerging evidence that patients with intermediate-risk PCa are a heterogeneous group, thus leading to a possible further stratification into two different prognostic groups (FG and UG) with different cancer-specific survival, biochemical relapse, and local and distant recurrence. This sub-classification, not included in the classical three risk-group stratification of PCa, may simplify treatment recommendations and lead to personalized treatment approaches [16,17,18,24,25,26].

In 2013, Zumsteg et al., investigated the survival outcomes of 1024 intermediate-risk PCa patients treated with dose-escalated RT (≥81 Gy) with or without HT (median duration of 6 months), stratifying the results for FG and UG. All risk stratification factors for intermediate-risk PCa were highly significant predictors of poor outcomes and, consequently, FG and UG have markedly different prognoses [14]. The authors confirmed these results and validated this modified risk stratification system in a sequent analysis on 2705 patients, suggesting that FG and low-risk PCa have comparable outcomes, as do UG and high-risk PCa [25]. Tom et al., retrospectively reported the outcomes of 1510 patients treated with definitive low-dose rate interstitial brachytherapy with or without HT (median duration of 6 months). They found significantly higher rates of biochemical failure and distant metastasis among men with UG compared with those with FG, thus supporting the subgroup risk classification and suggesting that UG could benefit from treatment intensification [26].

In our study, we did not find a different outcome between FG and UG patients. Only cPFS was statistically different in both univariate and multivariate analysis (*p* = 0.042 and *p* = 0.047, respectively), although, after PS matching, these results were not confirmed. This difference may be related to the relatively low number of cancer-related events and/or to the efficacy of the intensified treatment used being biologically equivalent to 80.5 Gy in 2 Gy fractions assuming a α/β ratio of 1.5 Gy. Moreover, in the definition of FG and UG, we have not considered the percentage of positive biopsies because there data are missing in a significant portion of patients.

We reported, in a previous paper, the potential role of PSA at first follow-up (i.e., after 6 months of HT) as a predictive factor for bPFS, MFS, and LF [27]. The results from an actual study with a longer follow-up and larger sample size of patients validate the previous hypothesis. PSA at first follow-up <0.8 ng/mL was significantly related to better oncological outcomes (CSS, bRFS, LRFS, cPFS, and MFS) in both univariate and multivariate analysis. Awaiting further refined risk classifications (e.g., clinical-genomic risk classifications), first follow-up PSA after RT may represent another useful and practical clinical tool that may help clinicians to personalize intermediate-risk PCa treatments. According to this value, clinicians may consider some options such as prolonging HT, shifting towards long-course adjuvant HT, or setting close follow-up in order to early detect loco-regional recurrences or systemic progressions.

The use of different image-guided radiotherapy strategies in PCa ablative RT treatments is part of daily clinical practice [28]. Those techniques help radiation oncologists to improve daily set-up, deliver high conformal doses, reduce PTV margins, decrease dose distributions to OARs, and thus reduce potential acute and late toxicities. We investigate, in this study, the impact of IGRT, showing no statistically significant differences in oncological outcomes for patients treated with or without daily cone-beam CT IGRT. These results may be explained with the different PTV margins adopted for IGRT versus no IGRT, ensuring an adequate tumor dose coverage in both approaches. In our cohort of patients, the reduction in PTV margins and the use of cone-beam CT IGRT did not lead to a statistically significant difference in acute and late toxicity profiles. This could be because of the intrinsic good tolerance of the treatment schedule used.

## 6. Conclusions

In conclusion, our study demonstrates the efficacy and tolerability of the schedule used (3.65 Gy for 15 fractions 3/w) regardless of the use of IGRT in patients in both the favorable and unfavorable prognostic group. This leads to a notable reduction in the number of hospital accesses by patients.

PSA at first follow-up is the only prognostic factor and can be an easily accessible predictive tool that may help clinicians to identify patients who require a treatment intensification.

## 7. Patents

We report the results and tolerability of a 15-fraction hypo-fractionated schedule in patients with intermediate-risk prostate cancer. We found that the treatment was efficacy and well tolerated, reducing the number of hospital accesses compared to a standard treatment. PSA value at first follow-up is an important prognostic factor.

## Figures and Tables

**Figure 1 jcm-11-04783-f001:**
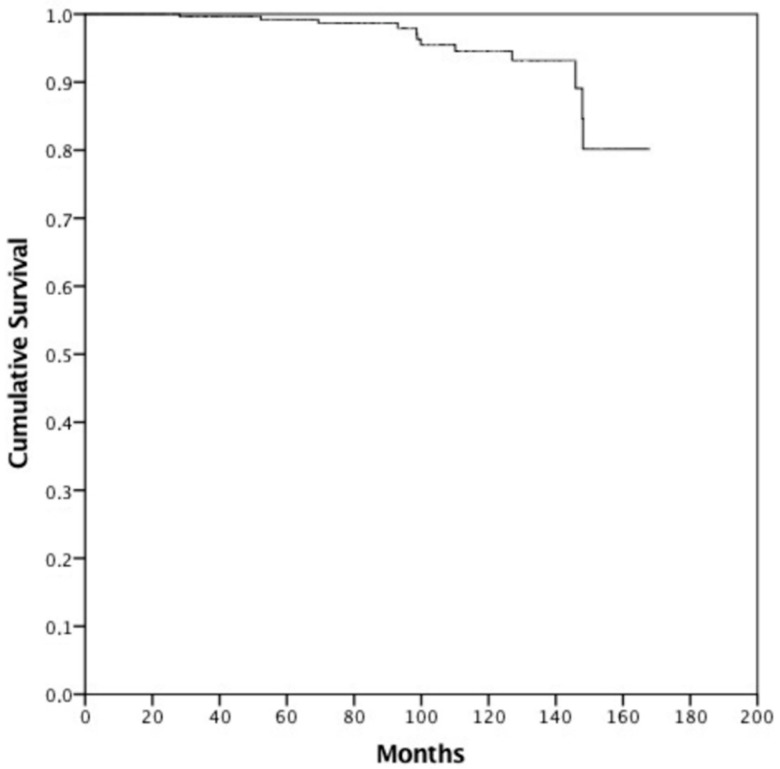
Cancer-specific survival.

**Figure 2 jcm-11-04783-f002:**
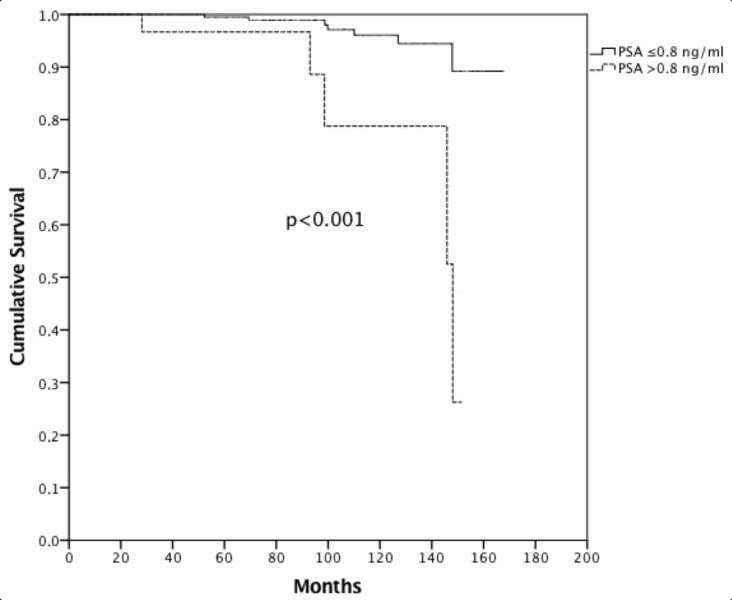
Cancer-specific survival in patients with PSA ≤ 0.8 ng/mL at first follow-up vs. >0.8 ng/mL.

**Table 1 jcm-11-04783-t001:** Patients’ characteristics.

*Total Number of Patients: 313*	
*Age*	
Median (range)	*74 years (48–88)*
≤70 years	58 pz (18.5%)
>70 years	255 pz (81.5%)
*PSA at diagnosis*	
Median (range)	*8.15 ng/mL (2.6–19.9)*
<10 ng/mL	198 *(63.2%)*
10–9.9 ng/mL	115 *(36.3%)*
*Tumor Stage*	
T1c	68 *(21.8%)*
T2a	92 *(29.4%)*
T2b	78 *(24.9%)*
*T2c*	*75 (23.9%)*
*Gleason Score*	
3 + 3	91 *(29.1%)*
3 + 4	133 *(42.5%)*
4 + 3	89 *(28.4%)*
*Hormonal Therapy*	
Antiandrogen	97 (31.0%)
LHRH analogue	216 *(69.0%)*
*Risk Group*	
Favorable Group	189 (60.4%)
Unfavorable Group	124 (39.6%)

LHRH: luteinizing hormone-releasing hormone, pz: patients.

**Table 2 jcm-11-04783-t002:** Patient and tumor characteristics before and after propensity score matching.

Variables	*Before Matching*			*After Matching*		
	Without IGRT *136 pz (43.5%)*	With IGRT *177 pz* *(56.5%)*	*Chi-Square* *(p-value* *)*	Without IGRT *112 pz*	With IGRT *112 pz*	*Chi-Square* *(p-value* *)*
**Age**			** *0.035* **			*0.304*
≤70 years	18 (5.8%)	40 (12.8%)		18 (8.0%)	24 (10.7%)	
>70 years	118 (37.7%)	137 (43.8%)		94 (42.0%)	88 (39.3%)	
**PSA at diagnosis**			*0.158*			*0.211*
<10 ng/mL	92 (29.4%)	106 (33.9%)		76 (33.9%)	67 (29.9%)	
10–19.9 ng/mL	44 (14.1%)	71 (22.7%)		36 (16.1%)	45 (20.1%)	
**Gleason Score**			*0.059*			*0.698*
3 + 3	49 (15.7%)	42 (13.4%)		25 (11.2%)	26 (11.6%)	
3 + 4	52 (16.6%)	81 (25.9%)		52 (23.2%)	46 (20.5%)	
4 + 3	35 (11.2%)	54 (17.3%)		35 (15.6%)	40 (17.9%)	
**Tumor Stage**			*0.422*			*0.592*
T1c–T2a	66 (21.1%)	94 (30.0%)		58 (25.9%)	62 (27.7%)	
T2b–T2c	70 (22.4%)	83 (26.5%)		54 (24.1%)	50 (22.3%)	
**Hormonal Therapy**			*<0.001*			*0.426*
Antiandrogen	23 (7.3%)	74 (23.6%)		23 (10.3%)	28 (12.5%)	
LHRH Analogous	113 (36.1%)	103 (32.9%)		89 (39.7%)	84 (37.5%)	

IGRT: image-guided radiation therapy, LHRH: luteinizing hormone-releasing hormone, pz: patients; PSA: prostatic specific antigen.

**Table 3 jcm-11-04783-t003:** Univariate analysis before and after propensity score matching.

Variables	*Before Matching*		*After Matching*		*Before Matching*		*After Matching*		*Before Matching*		*After Matching*	
	**8-year OS** **(N. Patients)**	*(p-value)*	**8-year OS** **(N. Patients)**	*(p-value)*	**8-year CSS**	*(p-value)*	**8-year CSS**	*(p-value)*	**8-year bRFS**	*(p-value)*	**8-year bRFS**	*(p-value)*
**IGRT**		*0.140*		*0.223*		*0.361*		*0.407*		*0.581*		*0.249*
Yes	85.6% (177)		82.9% (112)		97.1%		99.0%		86.0%		87.4%	
No	76.6% (136)		77.1% (112)		92.2%		97.0%		86.2%		84.5%	
**Age**		*0.557*		*0.923*		*0.187*	96.2%	*0.128*		*0.129*		*0.125*
≤70 years	84.6% (58)		81.0% (42)		97.3%		98.6%		81.4%		76.7%	
>70 years	81.3% (255)		79.7% (182)		98.0%				86.9%		88.5%	
**Favorable Group**	84.6% (189)	*0.841*	80.8% (127)	*0.808*	99.1%	*0.962*	98.6%	*0.676*	88.4%	*0.068*	83.3%	*0.715*
**Unfavorable Group**	77.3% (124)		78.7% (97)		95.8%		97.2%		81.6%		90.9%	
**Hormonal Therapy**		*0.270*		*0.958*		*0.078*		*0.205*		*0.917*		*0.531*
Antiandrogen	88.0% (97)		81.1% (51)		100.0%		100.0%		89.1%		93.1%	
LHRH Analogous	77.0% (216)		78.9% (173)		96.3%		97.3%		84.2%		83.6%	
**PSA before RT**		*0.715*		*0.937*		*0.145*		*0.206*		*0.237*		*0.306*
≤0.7 ng/mL	77.6% (83)		72.4% (66)		97.9%		97.2%		87.8%		88.3%	
>0.7 ng/mL	83.4% (230)		82.8% (158)		97.9%		98.4%		85.2%		85.2%	
**PSA at first FU**		** *0.019* **		** *0.001* **		** *<0.001* **	9	** *0.001* **		** *<0.001* **		** *0.011* **
≤0.8 ng/mL	83.7% (282)		82.2% (201)		98.9%		8.4%		88.1%		87.3%	
>0.8 ng/mL	65.6% (31)		60.9% (23)		88.6%		95.7%		65.9%		77.3%	
	**8-year LRFS**	*(p-value)*	**8-year LRFS**	*(p-value)*	**8-year c-PFS**	*(p-value)*	**8-year c-PFS**	*(p-value)*	**8-year MFS**	*(p-value)*	**8-year MFS**	*(p-value)*
**IGRT**		*0.275*		*0.467*		*0.443*		*0.835*		*0.693*		*0.831*
Yes	88.9%		87.7%		86.2%		97.7%		93.7%		94.4%	
No	95.9%		95.1%		94.4%		93.3%		97.7%		97.2%	
**Age**		*0.657*		*0.842*		*0.750*		*0.868*	93.	*0.447*		*0.891*
≤70 years	92.5%		89.3%		89.0%		87.0%		7%		93.7%	
>70 years	89.8%		92.0%		89.4%		91.5%		95.5%		96.4%	
**Favorable Group**	93.7%	*0.078*	90.3%	*0.818*	93.1%	*0.042*	89.5%	*0.884*	96.5%	*0.610*	94.6%	*0.259*
**Unfavorable Group**	84.7%		94.0%		83.1%		93.0%		93.1%		97.9%	
**Hormonal Therapy**		*0.673*		*0.935*		*0.659*		*0.868*		*0.862*		*0.605*
Antiandrogen	90.0%		92.9%		88.9%		92.9%		97.5%		93.1%	
LHRH Analogous	90.9%		91.0%		90.0%		89.8%		93.7%		94.3%	
**PSA before RT**		*0.948*		*0.798*		*0.665*		*0.940*		*0.691*		*0.735*
≤0.7 ng/mL	88.4%		89.1%		88.4%		89.1%		94.2%		92.4%	
>0.7 ng/mL	91.2%		92.6%		89.8%		91.3%		95.6%		97.3%	
**PSA at first FU**		** *0.014* **		*0.103*		** *0.005* **		** *0.026* **		** *0.004* **		** *0.003* **
≤0.8 ng/mL	91.8%		92.0%		91.0%		91.5%		96.8%		97.3%	
>0.8 ng/mL	76.3%		83.7%		73.7%		79.9%		78.8%		79.9%	

IGRT: image-guided radiation therapy; LHRH: luteinizing hormone-releasing hormone; PSA: prostatic specific antigen; RT: radiation therapy; FU: follow-up; OS: overall survival, CSS: cancer-specific survival; bRFS: biochemical relapse-free survival; LPFS: local progression-free survival, c-PFS: clinical progression-free survival; MFS: metastasis-free survival.

**Table 4 jcm-11-04783-t004:** Multivariate analysis before and after propensity score matching.

	*Before Matching*			*After Matching*		
OS	HR	95% CI	*(p-Value)*	HR	95% CI	*(p-Value)*
IGRT (Yes vs. No)	1.368	0.799–2.344	0.254	1.648	0.842–3.227	0.145
Age (≤70 years vs. >70 years)	0.923	0.479–1.777	0.810	1.178	0.556–2.494	0.669
FG vs. UG	0.973	0.587–1.615	0.917	1.032	0.561–1.898	0.920
HT (A vs. LHRHA)	0.842	0.484–1.465	0.544	1.112	0.584–2.119	0.746
PSA before RT (≤0.7 vs. >0.7)	1342	0.689–2.617	0.387	1.352	0.662–2.762	0.408
PSA at first FU (≤0.8 vs. >0.8)	0.468	0.243–0.900	0.023	0.307	0.147–0.641	0.002
CSS						
IGRT (Yes vs. No)	1.108	0.248–4.944	0.893	1.356	0.236–7.781	0.732
Age (≤70 years vs. >70 years)	2.580	0.788–8.454	0.117	2.090	0.560–7.797	0.272
FG vs. UG	0.699	0.205–2.380	0.567	0.797	0.203–3.125	0.745
HT (A vs. LHRHA)	0.431	0.094–1.977	0.279	0.623	0.117–3.324	0.580
PSA before RT (≤0.7 vs. >0.7)				0.530	0.058–4.880	0.575
PSA at first FU (≤0.8 vs. >0.8)	0.155	0.046–0.530	0.003	0.195	0.046–0.833	0.027
bPFS						
IGRT (Yes vs. No)	1.358	0.695–2.653	0.370	1.684	0.724–3.754	0.234
Age (≤70 years vs. >70 years)	2.154	1.073–4.326	0.031	2.192	0.966–4.971	0.060
FG vs. UG	0.541	0.291–1.004	0.052	0.986	0.461–2.111	0.972
HT (A vs. LHRHA)	0.929	0.466–1.853	0.835	0.730	0.304–1.753	0.481
PSA before RT (≤0.7 vs. >0.7)	1.158	0.488–2.745	0.740	0.902	0.335–2.432	0.839
PSA at first FU (≤0.8 vs. >0.8)	0.222	0.107–0.460	<0.001	0.283	0.111–0.721	0.008
LPFS						
IGRT (Yes vs. No)	0.624	0.241–1.612	0.330	0.729	0.252–2.113	0.561
Age (≤70 years vs. >70 years)	0.860	0.278–2.574	0.788	0.909	0.250–3.302	0.885
FG vs. UG	0.504	0.221–1.149	0.103	1.107	0.396–3.097	0.846
HT (A vs. LHRHA)	1.031	0.422–2.520	0.946	1.019	0.342–3.034	0.973
PSA before RT (≤0.7 vs. >0.7)	1.077	0.357–3.250	0.896	1.139	0.358–3.621	0.826
PSA at first FU (≤0.8 vs. >0.8)	0.328	0.120–0.894	0.029	0.368	0.100–1.354	0.113
c-PFS						
IGRT (Yes vs. No)	0.781	0.328–1.863	0.578	0.949	0.348–2.596	0.918
Age (≤70 years vs. >70 years)	1.340	0.528–3.400	0.538	1.197	0.382–3.750	0.757
FG vs. UG	0.454	0.208–0.989	0.047	0.997	0.381–2.612	0.996
HT (A vs. LHRHA)	1.065	0.457–2.479	0.885	0.897	0.310–2.596	0.841
PSA before RT (≤0.7 vs. >0.7)				1.072	0.343–3.345	0.905
PSA at first FU (≤0.8 vs. >0.8)	0.299	0.119–0.752	0.010	0.293	0.093–0.928	0.037
MFS						
IGRT (Yes vs. No)	0.776	0.236–2.551	0.676	1.502	0.332–6.793	0.597
Age (≤70 years vs. >70 years)	1.826	0.556–5.998	0.321	1.314	0.261–6.614	0.741
FG vs. UG	0.739	0.252–2.167	0.582	2.177	0.439–10.803	0.341
HT (A vs. LHRHA)	0.747	0.227–2.463	0.632	0.753	0.150–3.769	0.730
PSA before RT (≤0.7 vs. >0.7)	0.876	0.161–4.772	0.878	1.969	0.388–9.991	0.414
PSA at first FU (≤0.8 vs. >0.8)	0.204	0.063–0.661	0.008	0.149	0.034–0.641	0.011

HR: Hazard Ratio; IGRT: Image guided radiation Therapy; FG: Favorable group; UF: Unfavorable group; HT: Hormonal Therapy; A: Antiandrogen; LHRH: luteinizing hormone-releasing hormone; PSA: Prostatic specific antigen; RT: Radiation Therapy; FU: Follow-up; OS: overall survival, CSS: Cancer specific survival; bRFS: biochemical relapse free survival; LPFS: local progression free survival, c-PFS: Clinical progression free survival; MFS: metastasis free survival.

## Data Availability

Not applicable.

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
