# Peer review of "Hypofractionated Radiotherapy in Intermediate-Risk Prostate Cancer Patients: Long-Term Results"

_jcm, 2022, doi:10.3390/jcm11164783_

Round 1
Reviewer 1 Report
Manuscript. Entitled "Hypo fractionated radiotherapy in intermediate-risk prostate cancer patients: long-term result.", authors focus to evaluate survival and toxicities outcomes in a series of patients with intermediate-risk prostate cancer (PCa) treated with hypo fractionated radiotherapy (HyRT) + androgen deprivation therapy (ADT) with or without image guided radiotherapy (IGRT) and to investigate the impact of different variables. The overall strategies, methodology, and analyses well‐reasoned and appropriate to accomplish the specific aims of the study. Authors should consider some points in order to improve the manuscript quality:
1. While the study appears to be sound, the language is unclear, making it difficult to follow. Please go over your manuscript text and ensure it is written in an acceptable English language. There are many typing errors in the manuscript. I advise the authors work with a writing coach or copyeditor to improve the flow and readability of the text.
2. Abstract looking fine
3. Authors did not discussed the method of educating the patients for stimulation of empty the rectum and filled the bladder.
4. Please remove the methods explained in the last paragraph of the introduction
5. Add sound reason why you conduct this study in the last paragraph of introduction
6. Authors did not discussed the age and clinical stages wise toxicity level, what is the efficacy of IGRT in different clinical stages of diseases.
7. What is the difference in overall survival rate for HyRT +ADT +IGRT users and HyRT +ADT users
8. Please discussed the impact of dose heterogeneity on late normal tissue complication risk after hypo fractionated therapy
9. Authors claim that they did not find a different outcome between FG and UG patients, what is the reason of behind this. While many studies shown the differences in FG and UP patients.
10. Authors must discussed the strengths of this study, as a large number of studies have been published on Hypo fractionated radiotherapy in intermediate-risk prostate cancer patients: How this study is unique and different from other studies?
11. Is there any pre-assessment of androgen and GnRH agonist therapies on prostate cancer patients, what is the impact of hormonal therapy (androgen, GnRH) on IGRT level
12. How the patient positioned was verified in non IGRT group
Author Response
We want to thank the reviewer for his comments:
1) We have revised the English language;
2) Thanks for the comment
3) We have modified the sentence in the materials and methods as follows “Before planning CT scan, patients were asked to self-administer a mini enema to empty the rectum. Additionally, they were instructed to empty bladder and drink 500 ml of water about 30 minutes prior to the procedure”.
4) We have removed the last paragraph of the introduction;
5) We have inserted the following sentence “The aim of this study was to evaluate efficacy, toxicities and prognostic factors in a series of intermediate-risk prostate cancer patients treated with 15 fractions HyRT + ADT with or without image guided radiotherapy (IGRT)”.
6) We did not analyze the toxicity level in relation to age and stage for the low number of high grade toxicities recorded. We have not separately reported the data regarding the efficacy of IGRT based on the clinical stage in order not to burden the work with too much data and by incorporating stage, gleason and initial PSA in the definition of favorable or unfavorable group.
7) The results in terms of overall survival in patients treated with or without IGRT are shown in table 3 considering that all patients underwent HyRT plus hormone therapy.
8) In our study, the planned dose was sufficiently homogeneous and we inserted the following sentence in the toxicity section of the results: “The dose constraints for the rectum and bladder were respected in all patients and the dose to the targets varies between 95% and 107% of the prescription dose”.
9) We have explained the possible reasons by inserting the following sentence in the discussion: Those data may be related to the relatively low number of cancer-related events and/or to efficacy of the intensified treatment used biologically equivalent to 80.5 Gy in 2 Gy fractions assuming a α/β ratio of 1.5 Gy. Moreover, in the definition of FG and UG we have not considered the percentage of positive biopsies because it is a missing data in an important portion of patients.
10) We have added the following sentence in the discussion: The peculiarity of our study is the use of a hypofractionated schedule with 15 fractions of 3.65 Gy 3 times a week and, to our knowledge, it is the only study in the literature that uses this treatment schedule and with a long follow-up.
11) There wasn’t any pre-assessment of androgen and GnRH agonist therapies on prostate cancer patients; the choice was based on the physician's preference. There was no impact of the type of hormone therapy used on the results.
12) We have explained this point with the following sentence: “Patient position was verified using an electronic portal-imaging device (EPID) in the non-IGRT group, while the IGRT group underwent daily CBCT”.
Reviewer 2 Report
I’ve read with interest this work on hypofractionated RT in intermediate-risk Pca patients. The clinical question is original and the statistical analysis appropriate. Discussion covers the relevant literature. Here are my minor concerns:
- Authors assess that they considered 313 consecutive patients with intermediate risk Pca, and indicate as including criteria T2b-2c, GS 7 and PSA 10-20 ng/mL. They should indicate which guideline they used to select patients and reference it line 60-61.
- Line 150: insert the % for the 52 and 12 Pca patients.
Author Response
We want to thank the reviewer for his comments:
- We have changed the sentence as follows: “Three hundred thirteen consecutive patients with intermediate risk prostate cancer by the definition of NCCN guidelines (T2b–T2c or Gleason Score 7 or pre-treatment PSA 10 to 20 ng/mL) were treated between January 2005 and December 2019”.
- Inserted
